# ERK1/2-SOX9/FOXL2 axis regulates ovarian steroidogenesis and favors the follicular–luteal transition

Vijay Simha Baddela[1], Marten Michaelis[1], Xuelian Tao[1], Dirk Koczan[2], Jens Vanselow[1]

**Estradiol and progesterone are the primary sex steroids produced by the ovary. Upon luteinizing hormone surge, estradiol-producing granulosa cells convert into progesterone-producing cells and eventually become large luteal cells of the corpus luteum. Signaling pathways and transcription factors involved in the cessation of estradiol and simultaneous stimulation of progesterone production in granulosa cells are not clearly understood. Here, we decipher that phosphorylated ERK1/2 regulates granulosa cell steroidogenesis by inhibiting estradiol and inducing progesterone production. Down-regulation of transcription factor FOXL2 and up-regulation of SOX9 by ERK underpin its differential steroidogenic function. Interestingly, the incidence of SOX9 is largely uncovered in ovarian cells and is found to regulate FOXL2 along with CYP19A1 and STAR genes, encoding rate-limiting enzymes of steroidogenesis, in cultured granulosa cells. We propose that the novel ERK1/2-SOX9/FOXL2 axis in granulosa cells is a critical regulator of ovarian steroidogenesis and may be considered when addressing pathophysiologies associated with inappropriate steroid production and infertility in humans and animals.**

## Introduction

Ovarian granulosa cells have evolutionarily conserved steroidogenic functions mainly controlled by the consecutive actions of pituitary follicle-stimulating hormone (FSH) and luteinizing hormone (LH). FSH induces granulosa cells' estradiol production and promotes antral follicular development. Increased estradiol concentrations exert a positive feedback loop by promoting the synthesis of androstenedione in theca cells needed for estradiol production by granulosa cells. LH inhibits estradiol synthesis and triggers the luteinization of granulosa cells and the production of progesterone, followed by ovulation and the formation of the corpus luteum (Fig 1A). Eventually, the luteinized granulosa cells become the large luteal cells of the corpus luteum and continue the

progesterone production, which is a necessary signal for maintaining uterine endometrium throughout gestation. However, if the female has not conceived, the corpus luteum undergoes regression to allow the next follicular wave. Insulin and insulin-like growth factor-1 potentiate, whereas inflammatory cytokines like interleukin-1 and tumor necrosis factor-$\alpha$ inhibit the gonadotropin stimulation of ovarian estradiol and progesterone biosynthesis (Schams et al, 2001; Devoto et al, 2009; Mani et al, 2010). Well-balanced and timely dynamics of estradiol and progesterone levels are vital for successful reproduction in females. In addition, normal brain functioning, healthy bones, and healthy skin also require appropriate levels of these sex steroids, especially estradiol. Therefore, the signaling pathways mediating the granulosa cells' steroidogenesis are of great significance because the imbalance of ovarian estradiol and progesterone production causes infertility and various health issues in females (Adeyemi et al, 2018; Archer et al, 2019; Doretto et al, 2020).

Extracellular signal-regulated kinases 1 and 2 (ERK1 and ERK2) belong to a family of structurally related kinases called MAPK. ERK1/2 (MAPK3/1) signaling depends on the phosphorylation of two upstream cascade proteins, rapidly accelerated fibrosarcoma kinase and MEK kinase (MAP2K). The three-tiered MAPK pathway integrates cell-surface signals and transmits them to diverse downstream proteins, including transcription factors and transport proteins, to direct cellular responses. Multiple studies in mice clearly established that ERK is essential for LH-induced ovulation (Fan et al, 2009; Siddappa et al, 2015). Recently, intrafollicular injection of pharmacological ERK inhibitor PD0325901 into preovulatory follicles resulted in a significant decrease in the ovulation rate in cattle (Schuermann et al, 2018), indicating the conserved role of ERK signaling in ovulation. It was shown that transcription factors C/EBPα and β (CCAAT/enhancer-binding proteins), EGR1 (early growth response factor 1), and YAP1 (yes-associated protein 1) were activated by MAPK3/1 signaling in granulosa cells upon LH surge via epidermal growth factor signaling (Russell et al, 2003; Fan et al, 2009, 2011; Godin et al, 2022).

Despite clear evidence in ovulation, the role of MAPK3/1 signaling in the differentiation of granulosa cells is not yet clearly established (Moore et al, 2001; Dewi et al, 2002; Mani et al, 2010).

---

[1]Institute of Reproductive Biology, Research Institute for Farm Animal Biology (FBN), Dummerstorf, Germany  [2]Institute of Immunology, University of Rostock, Rostock, Germany

Correspondence: baddel@fbn-dummerstorf.de

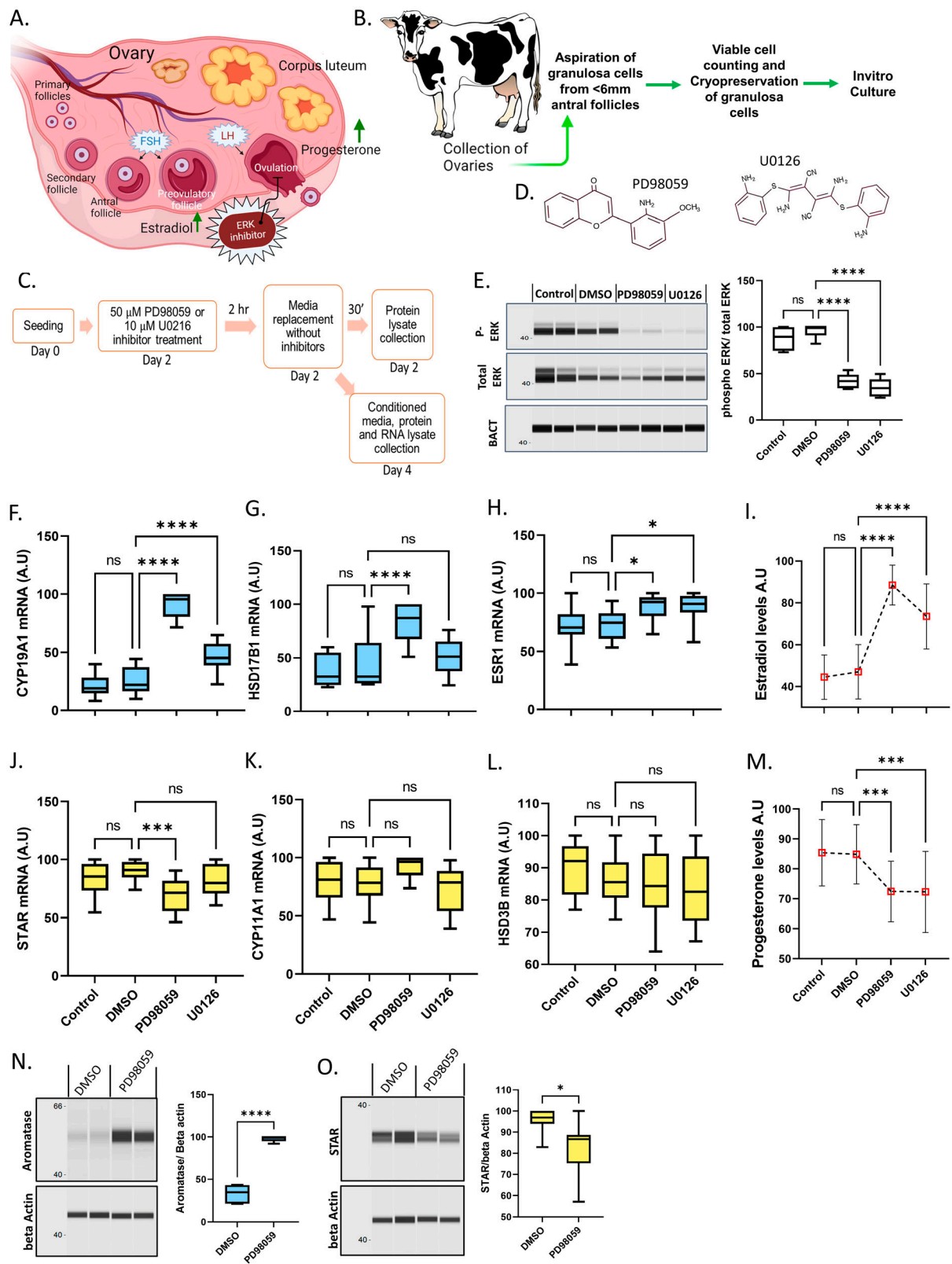

**Figure 1. ERK-induced regulation of estradiol and progesterone production.**
**(A)** Schematic representation of ovarian folliculogenesis wherein ERK inhibition halts ovulation. **(B)** Schematic representation of bovine ovarian granulosa cell collection and cryopreservation and culture. **(C)** Schematic representation of granulosa cell primary culture, treatments, and analysis time points for the experiments presented in the figure. **(D)** Structure of ERK1/2 inhibitors PD98059 and U0126. **(E)** Digitally constructed Western images of total ERK and phosphor-ERK in cultured cells in

Because follicular–luteal transition commences with estradiol to progesterone switching in granulosa cells of post-LH follicles (Fig 1A), here, we analyzed whether and how ovarian steroid hormone synthesis in granulosa cells is related to ERK activity. In contrast to the in vivo situation, primary granulosa cells under in vitro cell culture conditions produce both estradiol and progesterone, plausibly because of the lack of oocyte and thecal cell–derived factors such as growth differentiation factor-9 and bone morphogenetic protein (BMP) 15, BMP4, and BMP7 (Vanderhyden & Tonary, 1995; Spicer et al, 2006; Zhang et al, 2015; Rajesh et al, 2018). Here, we used bovine primary granulosa cells, which express all necessary steroidogenic genes (Baufeld & Vanselow, 2013; Baddela et al, 2022a, 2022b) and produce both estradiol and progesterone, to analyze the regulation of estradiol and progesterone production upon manipulating the ERK signaling.

# Results and Discussions

### Inhibition of MAPK3/1 phosphorylation induces estradiol and inhibits progesterone production

Collection of bovine ovaries, aspiration of follicles (<6 mm size), and primary culture of granulosa cells are depicted in Fig 1B and C. Chemical inhibitors PD98059 and U0126 are nonidentical molecules well known for blocking ERK phosphorylation via inhibiting MEK activity (Fig 1D). Based on earlier reports, 50 $\mu$M of PD98059 and 10 $\mu$M of U0126 were supplemented to the cultured granulosa cells to inhibit the ERK signaling (Cottom et al, 2003; Puttabyatappa et al, 2013; Donaubauer & Hunzicker-Dunn, 2016). Western analysis revealed that both inhibitors significantly decreased the ERK phosphorylation compared to control cells (Fig 1E), indicating the efficacy of ERK signaling inhibition under the present culture conditions. We analyzed three key genes such as CYP19A1, HSD17B1, and ESR1, related to estradiol synthesis and activity, and STAR, CYP11A1, and HSD3B, corresponding to progesterone synthesis to understand the regulation of steroidogenesis by ERK signaling. CYP19A1 encodes for aromatase, a rate-limiting enzyme of estradiol synthesis essential for converting androgens into estrogens in granulosa cells. HSD17B1 encodes 17β-hydroxysteroid dehydrogenase 1 that catalyzes estrone into estradiol in granulosa cells. Estrogen receptor alpha is a nuclear receptor encoded by the ESR1 gene that binds estradiol and exerts transcriptional activities. Various reports have shown that the estradiol–estrogen receptor system is vital for folliculogenesis (Liu et al, 2017). Present analyses revealed that both ERK inhibitors significantly up-regulated CYP19A1 expression (Fig 1F); however, the magnitude of induction caused by U0126 is less than that of PD98059. Expression of HSD17B1 is induced by PD98059 but not by U0126, whereas both inhibitors induced ESR1 expression (Fig 1G and H). The discrepancies in the effects caused by these inhibitors could

plausibly be because of the lower treatment concentration of U0126 (10 $\mu$M) compared with PD98059 (50 $\mu$M), thus, the low amount of positive transcriptional regulators for the expression of these genes by the given U0126 treatment. Ultrasensitive radioimmunoassay measurements of the conditioned media showed that both inhibitors have significantly increased estradiol production (Fig 1I). Despite the differences in the extent of the effect, the consistent induction of CYP19A1 expression and estradiol production by both inhibitors suggest that MAPK3/1 is a negative regulator of estradiol production in ovarian follicles.

On the other hand, CYP11A1 and HSD3B1 encode for cholesterol side-chain cleavage enzyme and β-hydroxysteroid dehydrogenase, respectively. These two enzymes sequentially catalyze the conversion of cholesterol into pregnenolone and then to progesterone in the mitochondria. STAR codes for the rate-limiting steroidogenic acute regulatory protein were involved in the import of cholesterol into mitochondria. The expression of CYP11A1 and HSD3B1 genes was unaffected by both inhibitors. The PD98059 treatment down-regulated STAR expression, but the U0126 treatment failed to regulate STAR expression (Fig 1J–L). We speculate that the discrepancy in STAR regulation by PD98059 and U0126, apart from the reasons mentioned in the above paragraph, could be because of the lower sensitivity of STAR gene expression to p-ERK as even the 50 $\mu$M PD98059 treatment showed only a slim down-regulation of STAR expression. Interestingly, despite no changes in gene expressions, U0126 caused a significant decrease in progesterone production, similar to the PD98059 treatment (Fig 1M). This is likely because of the inhibition of phosphorylation-driven activation of STAR protein by both inhibitors. It has been shown that STAR activity is increased by phosphorylation at serine 57 and 195 residues in COS-1 cells (Arakane et al, 1997). Studies in mouse and human granulosa cells have indicated that the phosphorylation-driven activation of STAR protein by MAPK3/1 induces cholesterol import into the mitochondria. Western analysis showed that similar to CYP19A1 and STAR mRNA data, aromatase expression is strongly up-regulated (Fig 1N), and the STAR protein is down-regulated (Fig 1O) by PD98059 treatment. These data suggest that by inhibiting estradiol production and inducing progesterone production in granulosa cells, MAPK3/1 signaling could act as a critical regulator of ovarian steroidogenesis in post-LH follicles and be necessary for follicular–luteal transformation.

### Knockdown-compelled ERK inactivation and TPA-driven ERK activation

It is of concern that the observed regulation of genes by PD98059 and U0126 may present some off-target effects (Wauson et al, 2013). Therefore the regulation of steroidogenesis by these inhibitors was validated using two different

---

the presence of ERK inhibitors PD98059 and U0123 and the quantification of data 30′ after media replacement (n = 5). **(F)** mRNA abundance of CYP19A1 gene (n = 12). **(G)** mRNA abundance of HSD17B1 gene (n = 12). **(H)** mRNA abundance of ESR1 gene (n = 12). **(I)** Radioimmunoassay quantification of estradiol levels in the culture media (n = 25). **(J)** mRNA abundance data of STAR gene (n = 12). **(K)** mRNA abundance data of CYP11A1 gene (n = 12). **(L)** mRNA abundance data of HSD3B gene (n = 12). **(M)** Radioimmunoassay quantification of progesterone levels in the culture media (n = 25). **(N)** Digitally constructed Western probing images of aromatase and beta-actin expression and their quantification (n = 6). **(O)** Digitally constructed western probing image of STAR and beta-actin protein expressions and their quantification (n = 7). Probability values < 0.05 were considered statistically significant and are designated with up to four asterisk symbols to inform the strength of significant difference (* = $P < 0.05$; ** = $P < 0.01$; *** = $P < 0.001$, **** = $P < 0.0001$). n = indicates the number of independent cell culture replicates analyzed.

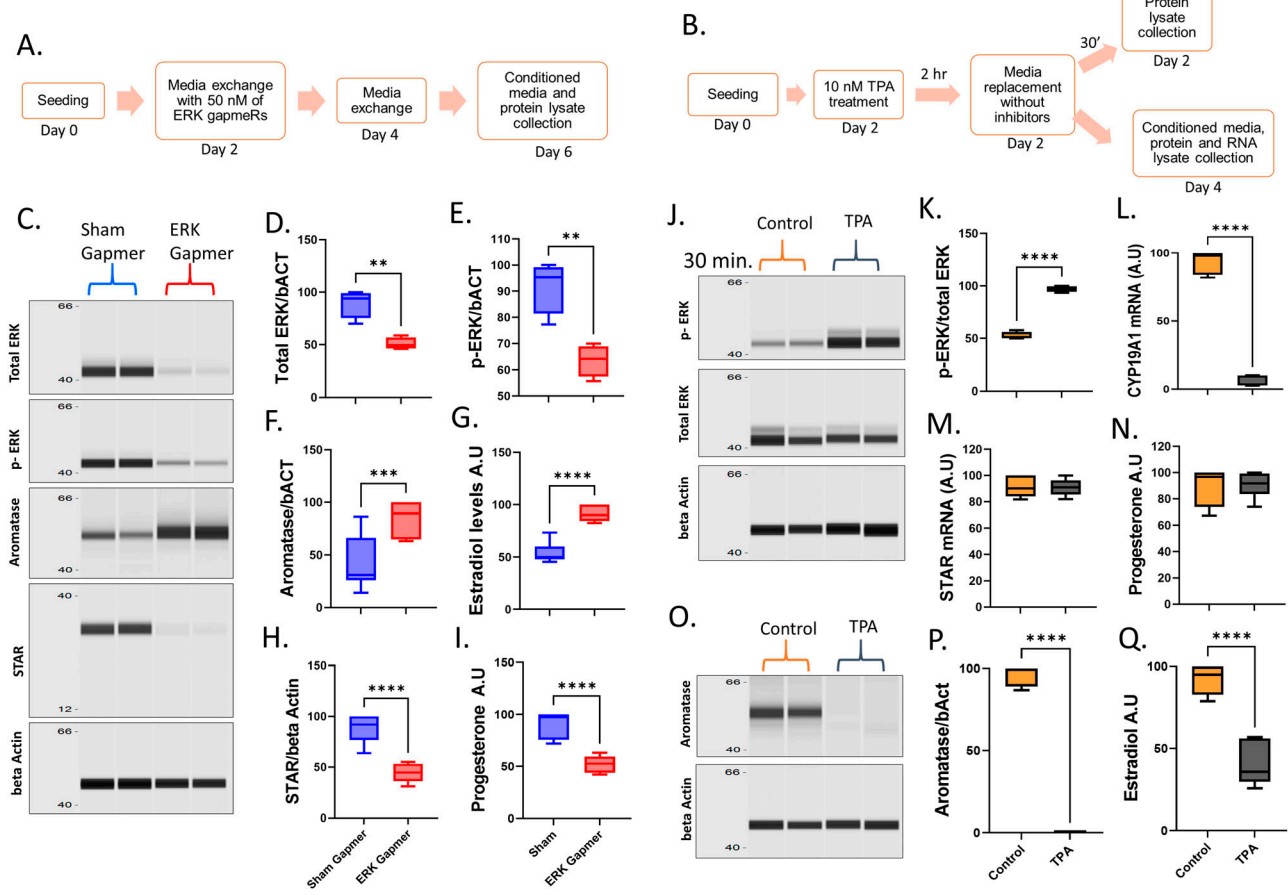

**Figure 2. ERK gene-silencing and ERK chemical activation in granulosa cells.**
**(A)** Schematic depiction of ERK-knockdown experiment. **(B)** Schematic depiction of the experiment for induction of ERK phosphorylation using TPA. **(C)** Digitally constructed Western probing images of total ERK, phospho-ERK, aromatase, STAR, and beta-actin proteins in granulosa cells upon ERK knockdown using antisense GapmeR technology. **(D)** Quantification of total ERK levels upon knockdown (n = 4). **(E)** Quantification of phospho ERK levels upon knockdown (n = 4). **(F)** Quantification of aromatase levels upon knockdown (n = 8). **(G)** Radioimmunoassay quantification of estradiol levels in the culture media (n = 8). **(H)** Quantification of STAR protein levels upon ERK knockdown (n = 8). **(I)** Radioimmunoassay quantification of progesterone levels in the culture media (n = 8). **(J)** Digitally constructed Western images of total ERK, phospho-ERK, and beta-actin proteins in cells treated with or without TPA. **(K)** Quantification of phospho ERK levels (n = 4). **(L)** mRNA abundance of *CYP19A1* gene (n = 6). **(M)** mRNA abundance data of *STAR* gene (n = 6). **(N)** Progesterone levels in the culture media (n = 6). **(O)** Digitally constructed Western images of aromatase and beta-actin proteins in cells treated with or without TPA. **(P)** Quantification of aromatase levels (n = 5). **(Q)** Radioimmunoassay quantification of estradiol levels in the culture media (n = 6). Probability values < 0.05 were considered statistically significant and are designated with up to four asterisk symbols to inform the strength of significant difference (* = $P < 0.05$; ** = $P < 0.01$; *** = $P < 0.001$, and **** = $P < 0.0001$). n = indicates the number of independent cell culture replicates analyzed.

approaches: (1) knockdown of ERK gene expression using antisense gapmeR technology (Fig 2A) and (2) activation of ERK signaling using TPA (12-O-Tetradecanoylphorbol-13-acetate) (Fig 2B) (Dhar et al, 2004; Wu et al, 2006). A significant decrease of the total and phospho ERK proteins upon ERK-specific antisense gapmeRs treatment indicated the efficacy of the ERK knockdown (Fig 2C–E). Granulosa cells showed dramatic up-regulation of aromatase protein and estradiol production (Fig 2C, F, and G) and down-regulation of STAR protein and progesterone production upon ERK knockdown (Fig 2C, H, and I). Interestingly, the level of decrease in STAR protein observed upon ERK knockdown was more significant than the inhibition of ERK phosphorylation, suggesting ERK proteins independent of their phosphorylation may also be involved in the induction of STAR gene expression, possibly by acting as a transcription cofactor (Fowler et al, 2011).

Contrary to the ERK knockdown, cells treated with TPA (10 nM) displayed robust up-regulation of ERK phosphorylation (Fig 2J and K) and nearly abolished *CYP19A1* expression (Fig 2L); however, *STAR* mRNA expression and progesterone production were found unchanged (Fig 2M and N). Aromatase protein (Fig 2O and P) and estradiol levels (Fig 2Q) were found in line with CYP19A1 mRNA (Fig 2L), which together provide clear evidence in support of the negative regulation of estradiol production by ERK phosphorylation. Overall, these data from different experimental approaches made it strikingly clear how MAPK3/1 signaling alone plays a vital role in regulating estradiol and progesterone production, which is absolutely critical for fertility in females. It has been shown that the granulosa lutein cells of the corpus luteum are bigger in size than the predecessor granulosa cells (Baddela et al, 2018). Because ERK signaling induces progesterone production and inhibits estradiol production in granulosa cells, which is analogous to luteal cells of

the corpus luteum, we wondered whether ERK also alters the cell size of granulosa cells. Analyses showed that modulation of ERK signaling did not alter the cell size (Fig S1), suggesting ERK signaling did not induce the complete transformation of granulosa cells into mature luteal cells.

Upon ovulation, granulosa cells display rapid proliferation and, together with other cell populations such as endothelial, fibroblast, and immune cells, form the solid tissue of the mature corpus luteum of up to 4–7 g or more from tissue remnants of an ovulated follicle in approximately 10 d in humans and cows (Zheng et al, 1994; Yoshioka et al, 2013). We, therefore, wondered whether ERK activation could also induce granulosa cell proliferation to support corpus luteum formation. We cultured granulosa cells and treated them with PD98059, TPA, and TPA+PD98059. Microphotographs of cells under different treatments indicated that ERK activation by TPA induced a significant increase in cell density, whereas TPA+PD98059 treatments showed cell density levels similar to the control group (Fig 3A, the upper blue color contrast images). Quantification of cell titer indicated that TPA significantly induced granulosa cell proliferation and cotreatment of PD98059 along with TPA inhibited the TPA-driven proliferation, indicating that granulosa cell proliferation was indeed induced by ERK signaling (Fig 3A, bar graphs). Color generation caused by cell titer reagent in different treatments can be seen next to corresponding bar graphs. Flow cytometer analysis of cellular DNA content under different treatment conditions showed that TPA caused a higher percentage of the cells in the G2/M phase and a lower percent of cells in the G0/G1 arrest phase of the cell cycle compared with other groups (Fig 3B; left side is the overlay of histograms and the right side is the quantification of cells in different stages of the cell cycle). PD98059 cotreatment prevented the TPA-induced cell cycle progression. Analysis of cell viability status by propidium iodide staining in a flow cytometer revealed no significant changes in viability by the ERK signaling modulators (Fig S2). These findings clearly show that MAPK3/1 induces granulosa cell proliferation that may be necessary for the mature corpus luteum formation.

## Genome-wide mRNA expression profile regulated by ERK signaling

Global gene expression analysis of granulosa cells treated or untreated with PD98059 was performed using GeneChip Bovine Gene 1.0 ST Arrays (Affymetrix, Inc.) and the data files were analyzed using the TAC 4.0 software. Transcriptome data distribution signals from individual files indicate the optimum quality of processed data (Fig S3). Principal component analysis (PCA) showed a clearly distinct distribution of samples of the control group and PD98059 group with 40% variance on the PCA1 axis (Fig 3C) followed by 17% and 15.4% variances in PCA2 and PCA3 axes, respectively, indicating the significant differences in the global gene expression profiles caused by ERK signaling. Expression values for 24,415 gene clusters were recorded in the transcriptome data (Table S1). Using the enrichment parameters false discovery rate (FDR) $P < 0.05$ and |FC| > 1.4, we identified that the expression of 654 gene clusters was differentially regulated, among which 220 were up-regulated and 434 were down-regulated by PD98059 (Fig 3D). Top 20 regulated genes in the dataset are presented as a heat map (Fig 3E).

Functional annotation analysis of differentially regulated genes was performed using an ingenuity pathway analyzer (IPA; QIAGEN). Canonical pathway annotations (Fig 3F) showed that ERK signaling is inhibited in the PD98059-treated cells, which can be considered as quality control for the inhibition of ERK phosphorylation by PD98059. The TGF-$\beta$ pathway, which was reported to inhibit the luteinization of granulosa cells (Zheng et al, 2009), was induced and regulation of epithelial to mesenchymal transition was inhibited in the PD98059-treated cells. Because follicular–luteal transformation is recognized as an epithelial to mesenchymal transition process (Abedal-Majed et al, 2019), these bioinformatics interpretations further support our hypothesis about ERK's role in the follicular–luteal transition. Upstream regulator analysis (Fig 3G) showed the significant enrichment of genes associated with FSH, LH, and IGF1 signaling in which FSH was activated (IPA Z score = +2.12; $P = 2.25 \times 10^{-10}$), whereas LH (IPA Z score = −1.51; $P = 1.98 \times 10^{-7}$) and IGF1 (IPA Z score = −2.32; $P = 1.18 \times 10^{-17}$) were inhibited in the PD98059-treated group. The full list of canonical pathways and upstream regulators controlled by ERK signalling can be found in Tables S2 and S3, respectively. Biological function analysis showed ovarian functions (Fig 3G) like estrus/menstrual cycle ($P = 4.78 \times 10^{-9}$), development of female reproductive tract ($P = 2.02 \times 10^{-8}$), quantity of antral follicles ($P = 6.62 \times 10^{-6}$), growth of the antral follicle ($P = 3.92 \times 10^{-5}$), and folliculogenesis (FLG; $P = 3.07 \times 10^{-5}$) were significantly enriched in the PD98059-treated cells. Together, all the above analyses clearly indicate that MAPK3/1 activation is essential for the preovulatory follicle development, ovulation, and the formation of the corpus luteum.

Transcriptome data also revealed that ERK inhibition by PD98059 induced mRNA abundance of FSH receptor (FSHR) and insulin-like peptide 3 (INSL3) genes and inhibited the PTX3 (Pentraxin-related protein 3) gene in granulosa cells (Table S1 and Fig S4). It was shown that LH down-regulates FSHR and up-regulates PTX3 expression in granulosa cells (Christenson et al, 2013). Therefore, the present data suggest that activation of ERK signaling by LH might also be responsible for the LH-induced regulation of FSHR and PTX3 genes in granulosa cells. Interestingly, INSL3, which was shown to associate with steroidogenic luteal cells (Balvers et al, 1998; Hanna et al, 2010; Satchell et al, 2013; Pitia et al, 2021), was also induced by PD98059 treatment, suggesting INSL3 gene may also regulate the steroidogenesis in granulosa cells (Fig S4).

## ERK-induced differential steroidogenesis is mediated by FOXL2 and SOX9

ERK modulates gene expression via activating transcription factors/cofactors. Therefore, we focused on identifying the possible transcriptional regulators mediating ERK effects in granulosa cells. Curation of transcriptome data revealed that forkhead box L2 (FOXL2) expression is inhibited (Fig 4A), and SRY-Box Transcription Factor 9 (SOX9) expression is induced (Fig 4B) by MAPK3/1 signaling. FOXL2 is a well-known positive regulator of female gonad development and a suppressor of SOX9 expression in female gonads. In addition, it is an activator for CYP19A1 gene transcription and, therefore, estradiol production (Wang et al, 2007; Uhlenhaut et al, 2009). Interestingly, FOXL2 mRNA expression was shown to be decreased in the granulosa cells of preovulatory follicles at 6 h post-hCG treatment. Furthermore, the newly formed CL was shown to have a complete loss of FOXL2

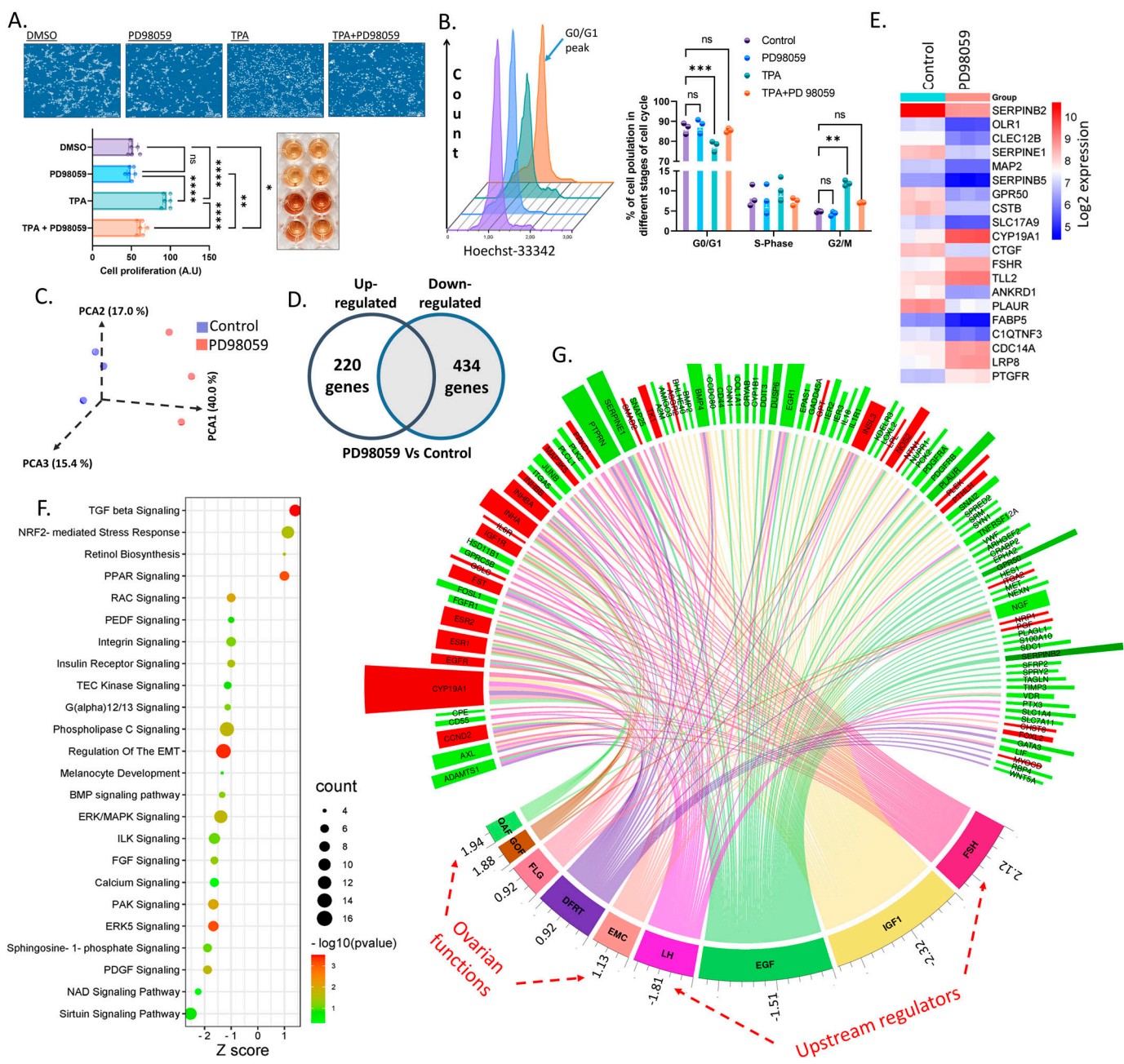

**Figure 3. ERK-induced changes in cell proliferation and transcriptome of granulosa cells.**

**(A)** Upper panel: blue contrast photomicrographs representing the cell density of granulosa cells cultured under different treatments. Bottom left: cell titer quantification data (n = 6). Bottom right: assay plate picture showing the color formation in treatments corresponding to the bars in the quantification data. Probability values < 0.05 were considered statistically significant and are designated with up to four asterisk symbols to inform the strength of significant difference (* = P < 0.05; ** = P < 0.01; *** = P < 0.001, and **** = P < 0.0001). n = indicates the number of independent cell culture replicates analyzed. **(B)** Left: overlayed histograms of granulosa cells in cell cycle analysis. Arrow indicates the G0/G1 peak. The colors of histograms correspond to the colors of bars in the right panel. Right: quantification of cells in different stages of the cell cycle (n = 3, each n is a pool of two replicates). **(C)** Principal component analysis of mRNA transcriptome data of control (blues) and PD98059-treated granulosa cells (reds). **(D)** Venn diagram representation of differentially expressed genes in the transcriptome data. **(E)** Heatmap of top 20 differentially expressed genes in the transcriptome data. **(F)** Functional annotation of enriched canonical pathways in PD98059-treated cells. The bottom x-axis indicates the IPA Z score, size of the bubble indicates the number of genes and the color of the bubble indicates the –log P-values. **(G)** Circos map of relevant upstream regulators and biological functions together with respective differentially regulated genes with expression indicators (red bar: up-regulated, green bar: down-regulated, height of the bar is proportional to the strength of significance in PD98059-treated cells). The numerical values below the annotation are the IPA Z-score indicating the activation or inhibition of annotation in PD98059-treated cells. FSH, follicle-stimulating hormone; LH, luteinizing hormone; IGF1, insulin-like growth factor 1; EGF, epidermal growth factor; EMC, estrus/menstrual cycle; DFRT, development of female reproductive; FLG, folliculogenesis; QAF, quantity of antral follicles; GOF, growth of the antral follicle.

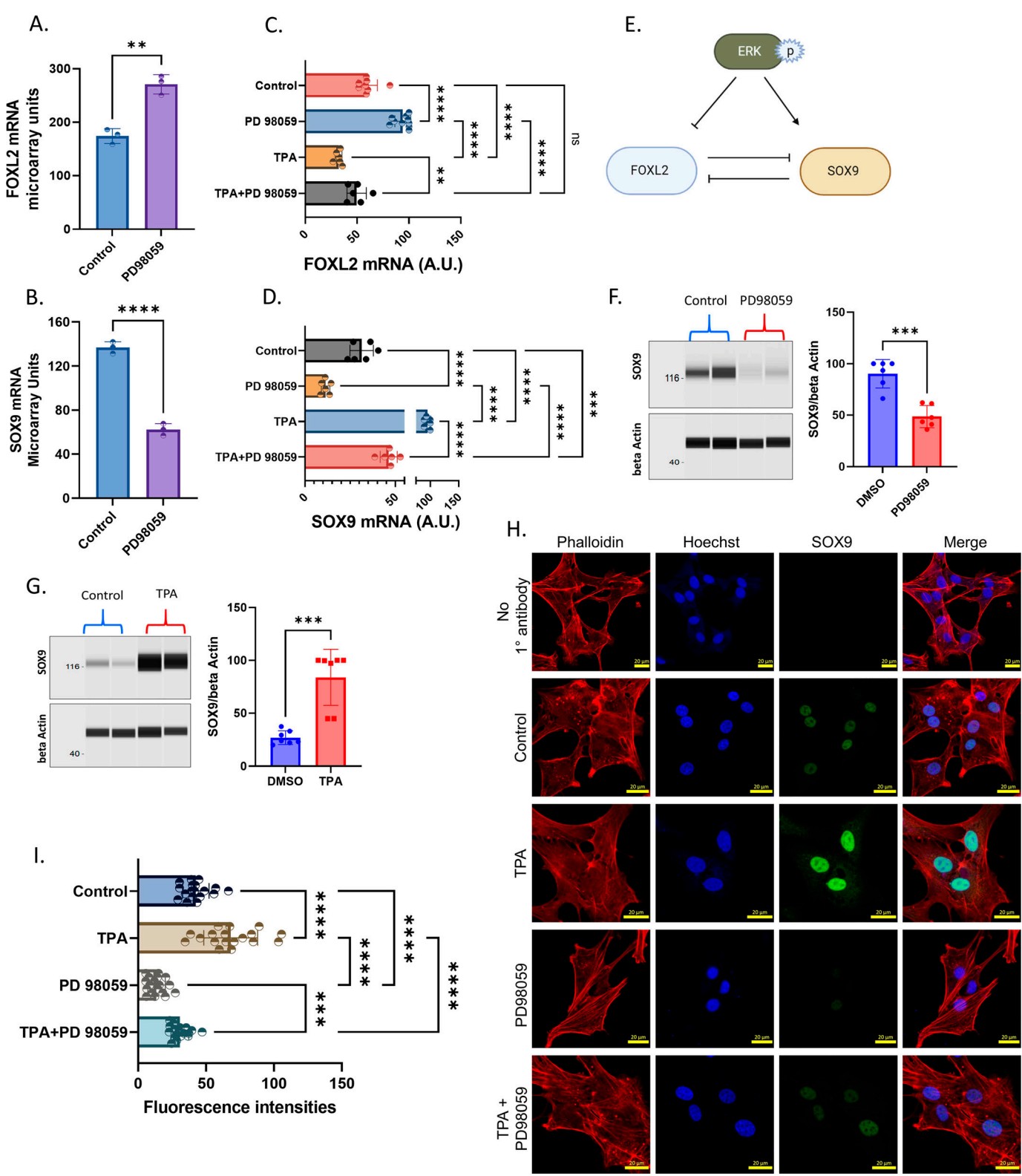

**Figure 4. ERK regulation of FOXL2 and SOX9 underpins the steroidogenic function.**
**(A)** Expression intensities of the *FOXL2* gene in microarray data (n = 3). **(B)** Expression intensities of the *SOX9* gene in microarray data (n = 3). **(C)** mRNA quantification of *FOXL2* in granulosa cells cultured under different conditions (n = 6). **(D)** mRNA quantification of *SOX9* in granulosa cells cultured under different conditions (n = 6). **(E)** Schematic representation of ERK's regulation of *FOXL2* and *SOX9* genes in granulosa cells and also possible bidirectional regulation of FOXL2 and SOX9 based on earlier reports. **(F)** Digitally constructed Western images of SOX9 and beta-actin proteins in control and PD98059-treated granulosa cells and their quantification data (n = 6).

expression at 16 h post-hCG, indicating FOXL2 is not essential for the luteinization process and may in fact act as a repressor of the luteinization of granulosa cells (Pisarska et al, 2004). This was confirmed by the constitutive expression analysis of the *FOXL2* in mouse ovarian somatic cells, which led to the loss of *STAR* expression and caused anovulation (Nicol et al, 2020). Therefore, ERK-driven decrease in estradiol production and increase in progesterone production observed in the present study could be a result of FOXL2 suppression by ERK. To clarify the inhibition of FOXL2 by ERK in granulosa cells, we treated cells with TPA, PD98059, and TPA+PD98059 and analyzed *FOXL2* mRNA expression. Results showed that activation of ERK using TPA inhibited the *FOXL2* expression; on the contrary, PD98059 induced *FOXL2* expression, as seen in transcriptome data. Cotreatment of cells with PD98059 along with TPA significantly reversed TPA's inhibition of *FOXL2* expression (Fig 4C), revealing that ERK inhibits the expression of *FOXL2*, thus eventually suppressing the *CYP19A1* gene expression and estradiol production. On the contrary, ERK induces *STAR* expression and progesterone production in granulosa cells.

On the other hand, recombinant LH treatment was found to induce *SOX9* mRNA expression in human primary granulosa cells derived from patients undergoing in vitro fertilization procedure (Lan et al, 2013). SOX9 was reported to inhibit the *FOXL2* transcription in mice and bovine fetal testes and plays a vital role in testicular development (Croft et al, 2018). Among nonreproductive tissues, cartilages require SOX9 activity for their development and function in both sexes. Interestingly, ERK was reported to induce *SOX9* expression in chondrocytes and UroCa cell lines (Murakami et al, 2000; Ling et al, 2011; Peacock et al, 2011), which is in line with our present microarray data in granulosa cells. However, to clarify this, we quantified *SOX9* mRNA expression under TPA, PD98059, and TPA+PD98059 treatments. Results showed that activation of ERK using TPA robustly increased *SOX9* expression; on the contrary, inhibition of ERK using PD98059 decreased *SOX9* expression compared with the control (Fig 4D), which indicated the opposite regulation of *FOXL2* and *SOX9* genes by ERK in granulosa cells (Fig 4E). Western analysis further confirmed these findings as SOX9 protein expression decreased in cells treated with PD98059 (Fig 4F) and increased in cells treated with TPA (Fig 4G) compared with respective controls. Immunofluorescence localization of SOX9 in granulosa cells (Fig 4H and I) indicated that SOX9 is mainly localized in the nucleus and the its fluorescence intensities are proportional to ERK phosphorylation levels and to that of *SOX9* mRNA and protein abundancies under different inhibitor treatments (Fig 4D, F, and G).

However, till now, no direct evidence has been reported on the role of SOX9 in the functioning of ovarian cells. Therefore, to understand the significance of SOX9, we conducted *SOX9* knockdown using antisense gapmerRs in cultured granulosa cells (Fig 5A). Western analysis showed significant down-regulation of SOX9 expression upon gapmeR treatment, indicating the efficiency of the knockdown procedure in TPA-treated cells (Fig 5B). The expression of *FOXL2* and *CYP19A1* genes and the estradiol production significantly increased upon SOX9 knockdown (Fig 5C–E). In contrast, the expression of *STAR* mRNA and progesterone production were down-regulated compared with the sham GapmeR treatment (Fig 5F and G). These results show that ERK-dependent inhibition and induction of estradiol and progesterone productions, respectively, are also plausibly mediated via up-regulating SOX9 expression. We then asked whether SOX9, being a transcriptional regulator, could regulate the expression of *FOXL2*, *CYP19A1*, and *STAR* by binding to their promoters. Upon transcription factor binding analysis using PROMO, a virtual tool for analyzing the putative transcription factor binding sites available at https://alggen.lsi.upc.es/recerca/frame-recerca.html, we identified plausible SOX9-binding sites having consensus with the reported SOX9-binding sequence (A/T)(A/T)CAA(A/T)G (Oh et al, 2010) in the putative promoter regions of *FOXL2*, *CYP19A1*, and *STAR* genes. Chromatin immunoprecipitation (ChIP) analysis of these putative SOX9-binding regions identified the enrichment of the SOX9 protein on the *FOXL2* and *STAR* genes but not on *CYP19A1* (Fig 5H). Together with *SOX9* knockdown data, these ChIP data suggest that SOX9 regulates the expression of *CYP19A1* and estradiol production indirectly via down-regulating *FOXL2* and increases progesterone production by directly activating *STAR* expression. Interestingly, a study on breast cancer cells showed that progesterone supplementation could inhibit aromatase expression, whereas inhibition of progesterone signaling by ablation of progesterone receptor induced the aromatase expression (Hardy et al, 2008), suggesting that such additional mechanisms involving steroid receptors could potentiate the ERK-SOX9/FOXL2-driven regulation of steroidogenesis in post-LH follicles and help the follicular–luteal transformation process.

Taken together, the present data on steroidogenic primary granulosa cells reveal that besides being essential for inducing the ovulation process, ERK signaling also regulates ovarian steroidogenesis and induces the proliferation of granulosa cells that may support the successful follicular luteal transformation. Importantly, ERK signaling down-regulates *FOXL2* and up-regulates *SOX9* expression, which eventually ceases *CYP19A1* expression and estradiol production, while increasing *STAR* and progesterone production (Fig 5I). Because optimum estradiol levels are also very important for maintaining healthy bones, skin, heart, and also brain function, we suggest that future studies are needed to determine whether ERK inhibitors can be used in humans for addressing low estrogen-driven disorders. Similarly, future studies need to be carried out on whether in vivo corpus luteum function can be improved by ERK activators to address luteal insufficiency, which is a major cause of early embryonic death in cows.

**(G)** Digitally constructed Western images of SOX9 and beta-actin proteins in control and TPA-treated granulosa cells and their quantification data (n = 7).
**(H)** Immunofluorescence images of SOX9 localization in granulosa cells under different treatment conditions. The scale bar is 20 μm. **(I)** Quantification of nuclear SOX9 signals from immunofluorescence images (Fig 4H) taken at different locations of the culture plate (n = 3). Probability values < 0.05 were considered statistically significant and are designated with up to four asterisk symbols to inform the strength of significant difference (* = *P* < 0.05; ** = *P* < 0.01; *** = *P* < 0.001, and **** = *P* < 0.0001). n = indicates the number of independent cell culture replicates analyzed.

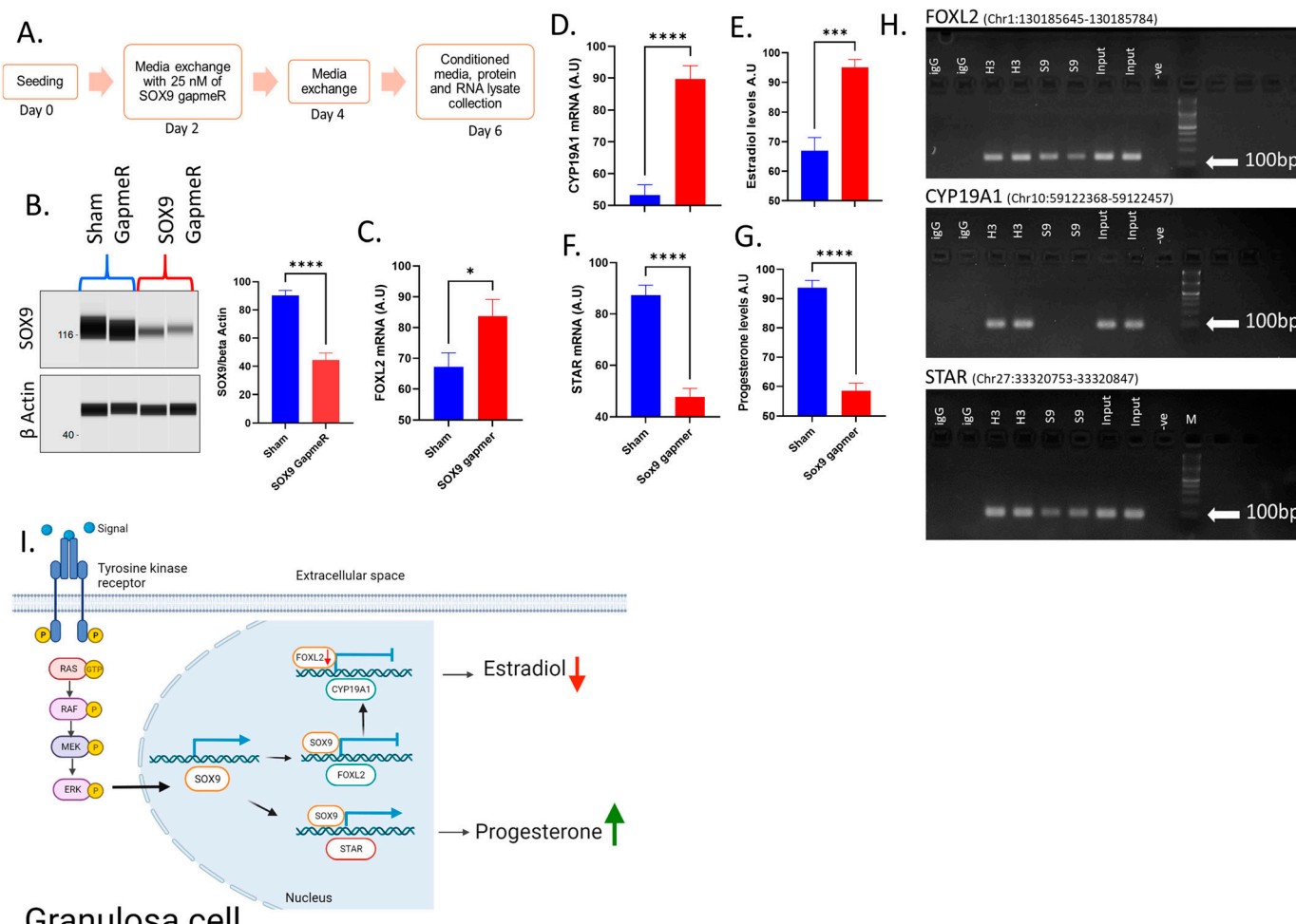

**Figure 5. SOX9-induced regulation of steroidogenesis.**
**(A)** Schematic depiction of *SOX9* gene knockdown experiment. **(B)** Digitally constructed Western images of SOX9 and beta-actin proteins after Sham and SOX9 knockdown in granulosa cells treated with TPA and their quantification data (n = 8). **(C)** mRNA quantification of *FOXL2* in granulosa cells upon *SOX9* knockdown (n = 14). **(D)** mRNA quantification of CYP19A1 in granulosa cells upon SOX9 knockdown (n = 14). **(E)** Radioimmunoassay quantification of estradiol levels in the culture media (n = 14). **(F)** mRNA quantification of *STAR* in granulosa cells upon *SOX9* knockdown (n = 14). **(G)** Radioimmunoassay quantification of progesterone in the culture media (n = 14). **(H)** 3% agarose gel images indicating SOX9 enrichment on *FOXL2*, *CYP19A1*, and *STAR* putative promoter regions in two samples. IgG: negative control antibody; H3: positive control histone H3 antibody, S9: SOX9 antibody, input: 2% input sample, -ve: PCR negative control. **(I)** Graphical conclusion of present data: induction of ERK phosphorylation in granulosa cells decreases the expression of FOXL2 and increases that of SOX9, and by doing so, ERK induces *STAR* and inhibits *CYP19A1* and therefore, progesterone and estradiol levels, respectively, in granulosa cells. Probability values < 0.05 were considered statistically significant and are designated with up to four asterisk symbols to inform the strength of significant difference (* = P < 0.05; ** = P < 0.01; *** = P < 0.001, and **** = P < 0.0001). n = indicates the number of independent cell culture replicates analyzed.

# Materials and Methods

### Granulosa cell culture model

For cell culture usage, $\alpha$-MEM (#F0915; Bio and Sell GmbH) was reconstituted by supplementing 0.1% BSA, 4 ng/ml sodium selenite, 5 $\mu$g/ml transferrin, 10 ng/ml insulin, 0.084% sodium bicarbonate, 20 mM HEPES, 2 mM L-Glutamin, 1 mM nonessential amino acids, 100 IU penicillin, and 0.1 mg/ml streptomycin. In addition, 2 $\mu$M androstenedione, IGFI, and FSH (Sigma-Aldrich) were added on the day of culture. Bovine ovaries were obtained from the DANISH CROWN Teterower Fleisch GmbH, Teterow, Germany, and transported to the laboratory in 1X PBS supplemented with 100 IU/ml

penicillin, 100 $\mu$g/ml streptomycin, and 0.5 $\mu$g/$\mu$l amphotericin. Granulosa cells were collected by manually aspirating the follicular fluid from small- and medium-sized ovarian follicles ($\leq$6 mm), which have high *FSHR* expression and very low LH receptor (*LHCGR*) expression, using a 3-ml syringe and 18G needle. Viable cells were counted via the trypan blue staining exclusion method and cryopreserved in freezing media (FCS containing 10% DMSO). At the time of seeding the cells, cryovials were thawed in a water bath at 37°C and washed using $\alpha$-MEM, and dissolved in reconstituted media. Cell cultures were performed in collagen-coated culture plates with a seeding density of 3 × 10⁵ cells/ml. For inhibiting ERK activity, cells were treated with two different chemical ERK inhibitors, PD98059 (50 $\mu$M) or U0126 (10 $\mu$M), in $\alpha$-MEM for 2 h on day 2 of

culture. 10 nM of TPA (12-O-Tetradecanoylphorbol-13-Acetate) was used for inducing the ERK phosphorylation. The media were replaced with fresh media without the inhibitor after 2 h and the culture continued until day 4, depending on the experiment. The gapmeR oligos used for the knockdown of ERK1/2 and SOX9 genes, along with negative control Sham gapmeR sequences, are listed in Table S4. 50 nM ERK1/2 gapmeR mix and 25 nM SOX9 gampeR concentrations were used for knocking down respective genes using TransIT-X2 Dynamic Delivery System (MIR6000; Mirus Bio) by following the manufacturer's recommendations. For knockdown experiments, the corresponding gene-specific oligos were added to the cells on day 2 of the culture and the media were replaced with fresh media on day 4 and continued the culture until day 6 for the analysis.

## Radioimmunoassay

The concentrations of estradiol and progesterone in the conditioned media were measured via ultra-sensitive competitive radioimmunoassays using custom-generated and purified rabbit antibodies. [2, 4, 6, 7-3H] 17$\beta$-Estradiol (GE Healthcare) and [1, 2, 6, 7-3H (N)] progesterone (PerkinElmer) were used as tracers for estradiol and progesterone estimation, respectively. The estradiol standard (E-8875; Sigma-Aldrich) and progesterone standards (P-0130; Sigma-Aldrich) were dissolved in ethanol and diluted in PBS. The range of the standard curve was chosen from 480 to 0 pg/ml via double dilution of standards. Separation of free and antibody-bound steroids was performed by the dextran-activated charcoal method. The levels of radioactivity were measured in a liquid scintillation counting system equipped with an integrated RIA-calculation program (TroCarb 2900 TR; PerkinElmer). The minimum detection limit was 3 pg/ml for estradiol and 7 pg/ml for progesterone. The intra- and inter-assay coefficients of variation for estradiol were 9.9% and 6.9%. These were 7.6% and 9.8% for progesterone. All samples and standards were assayed in duplicates, and the average of which was taken for the data analysis.

## mRNA quantification

Total RNA was isolated from cultured cells using the innuPREP RNA Mini Kit (Analytik Jena) and quantified through a NanoDrop 1000 spectrophotometer (Thermo Fisher Scientific). cDNA was prepared using the SensiFAST cDNA synthesis kit (Bioline). SensiFAST SYBR No-ROX reagent and gene-specific primers (Table S5) were used for gene expression analysis. For all genes except INSL3, the PCR product was cloned in the pGEM-T vector (Promega) and sequenced to verify the specificity of the primer pairs. Five dilutions of cloned and verified plasmids were used as standards and amplified with cDNA samples. *INSL3* relative quantification was performed using the previously published primer pair (Satchell et al, 2013). The PCR amplification was performed in duplicates in a total of 12 $\mu$l reaction mix volume using a Light Cycler 96 instrument (Roche). The PCR amplicons were verified for each run using melting curve analysis and agarose gel electrophoresis.

## Capillary Western probing

Protein quantification was performed using the capillary Western method with ProteinSimple's WES instrument according to the manufacturer's guidelines. Cells were cultured in 48-well plates and lysed using 50 $\mu$l of 1x MPER lysis buffer (Thermo Fisher Scientific). The lysate was centrifuged at 4°C at 12,000$g$ for 3 min to collect the protein supernatant. Protein concentrations were assayed using the micro BCA protein quantification method (Thermo Fisher Scientific). Protein samples, wash buffers and blocking reagents, primary and secondary antibodies, and chemiluminescent substrates were prepared and distributed into dedicated wells of the assay plate. The assay plates were loaded onto the instrument for protein separation in a 12–230 kD capillary separation module (SM-W001). The detection of protein bands was automated by the instrument. The list of primary antibodies used for the analysis is listed in Table S6. The anti-mouse (DM-002) and anti-rabbit (DM-001) secondary antibody modules were purchased from ProteinSimple Company.

## Cell titer analysis

Cell titer analysis was performed using tetrazolium compound-based CellTiter 96 AQueous (Promega) according to the manufacturer's recommendations. Granulosa cells were cultured in 96-well plates in 100 $\mu$l media. At the end of the experiment, 20 $\mu$l of CellTiter 96 Aqueous solution was pipetted into each well and incubated at 37°C for 1 h in a cell culture incubator. Absorbance was recorded at 490 nm using a plate reader instrument.

## Cell cycle analysis

Cells were cultured for 2 d under basal conditions before giving treatment of either PD98059, TPA, PD98059 + TPA or DMSO as vehicle control. After 24 h of treatment, cells were detached by incubation with Accutase (A6964; Sigma-Aldrich) for 20 min at 37°C. After two wash steps with 1x PBS, cells were incubated in the culture medium containing 5 $\mu$g/ml Hoechst33342 for 30 min at 37°C in the dark. Immediately after staining, cells were analyzed using a flow cytometer (MoFlo XDP; Beckman-Coulter) equipped with a 355 nm laser for optimal excitation of Hoechst-33342. DNA signal was quantified from single cells (10,000 counts) using the area and height parameters of Hoechst33342. The data were subsequently analyzed using the MultiCycle Tool of FCS Express 6 software.

## Cell viability analysis

Cultured granulosa cells were washed with PBS and detached from the culture plate by incubating them with Accutase (A6964; Sigma-Aldrich) for 20 min at 37°C. Cells were transferred into a 1.5-ml tube and washed twice with pre-warmed DMEM by centrifugation at 300$g$ for 5 min. Cells were resuspended in DMEM containing 5 $\mu$l of propidium iodide (0.5 mg/ml in PBS). The fluorescence signal was quantified from single cells (10,000 counts) by a flow cytometer (Gallios; Beckman-Coulter) and the data obtained were analyzed using the Kaluza Software (Beckman-Coulter).

## Transcriptome analysis

RNA samples from Control and PD98059 treated cells were subjected to microarray analysis using bovine gene 1.0 ST arrays (Affymetrix). The quality of RNA was verified by a Bioanalyzer instrument (Agilent Technologies). All the samples used for the microarray analysis showed a RIN value of 9.5 to 9.9, indicating negligible degradation of RNA in the samples. Amplification, labeling, and hybridization procedures of microarray were performed with Gene Chip Expression 3′ amplification one cycle target labeling and control reagents supplied by Affymetrix using the recommended protocol. Hybridization was done overnight in the Gene Chip R hybridization oven, and the expression was visualized using Affymetrix Gene Chip Scanner 3000. Raw data were processed in the expression console software (Affymetrix; V1.46) for normalization and background reduction. The gene-level summary was performed using the Robust Multichip Average method. The functional annotation of transcriptome data were carried out using IPA (QIAGEN).

## Immunofluorescence analysis of SOX9

Granulosa cells were cultured in chamber slides coated with collagen for the imaging analysis. The cells were washed three times with 1x PBS and fixed using 4% PFA for 4 h at 4°C after which the cells were washed in 1x PBS and then subjected to permeabilization using 0.3% Triton X for 10 min at RT. Cells were incubated in 5% BSA solution at RT for 30 min for blocking nonspecific antibody binding. Cells were incubated with SOX9 antibody (1:100) at 4°C overnight, followed by rinsing four times in 1X PBS and then incubation with Alexa fluor 647 goat anti-rabbit secondary antibody (1:200) and incubated in the dark at RT for 1 h. Cells were rinsed again with 1X PBS to omit any excess bound antibodies mixture. Then, SYBR Green (1:1,000) was added and incubated in the dark at RT for 20 min. After this, 2% PFA was added and again incubated at RT for 20 min. Finally, cells were covered with mounting media and kept at 4°C until visualized under a laser scanning microscope (LSM 800; Carl Zeiss) using a 40x oil objective lens.

## ChIP

To clarify whether SOX9 binds to the promoter regions of *FOXL2*, *CYP19A1*, and *STAR* genes, ChIP was performed using the SimpleChIP Enzymatic Chromatin IP Kit (#9003; Cellsignal Technology). Granulosa cells were cultured for two days under basal conditions, followed by treatment with TPA for another 2 d to increase the SOX9 abundance and collected for analysis. Cells were detached by incubation with Accutase for 20 min at 37°C and washed two times with 1x PBS. ChIP was performed as described by the manufacturer but with some modifications. For each IP, 2 million granulosa cells were used, and to digest the chromatin, we found that 500 gel units of micrococcal nuclease were sufficient. After stopping the digestion by adding 10 µl of 0.5 M EDTA, sonication was done in two sets of 20-s pulses using the ultrasonic processor Labsonic M (Sartorius AG). Complete lysis of the nuclei was checked by light microscopy. To check the chromatin digestion, an aliquot of each lysate was incubated with RNAse A for 30 min at 37°C followed by

Proteinase K for 2 h at 65°C. The lysate was then loaded on a 1% agarose gel and analyzed by gel electrophoresis. ChIP was performed overnight at 4°C on a rotator with 1 µg of a rabbit IgG to SOX9 antibody (#82630; Cell Signaling). As a positive and negative control, we used 1 µg of Histone H3 antibody (#4620; Cell Signaling), and 1 µg of an unspecific rabbit IgG antibody (#2729; Cell Signaling), respectively. Eluted DNA from SOX9, histone H3, and unspecific rabbit IgG precipitation were subjected to PCR amplification, and the product was electrophoresed in 3% agarose gel for detecting the enrichment of SOX9 on the putative promoter regions of *FOXL2*, *CYP19A1*, and *STAR* genes using specific primers mentioned in Table S5.

## Statistical analysis

Statistical analyses and data visualization were performed using the Graph pad Prism 9.0 licensed software. Normalized data values were used for the statistical analysis unless otherwise stated. Unpaired two-way $t$ tests were performed for comparison between the two groups. The remaining data were analyzed using one-way ANOVA. Pairwise multiple comparisons were executed using appropriate post hoc tests. Data are presented either as box plots with whiskers spanning a minimum to a maximum point of the data read or bar charts of mean with SEM values. Probability values < 0.05 were considered as statistically significant and are designated with up to four asterisk symbols to inform the strength of significant difference (* = $P < 0.05$; ** = $P < 0.01$; *** = $P < 0.001$, **** = $P < 0.0001$). The transcriptome data were analyzed using TAC 4.0 software (Affymetrix). Analysis of Variance (ANOVA) was used to calculate the $P$-value and was additionally corrected for FDR (Benjamin–Hochberg method) integrated within the TAC 4.0. Differentially expressed genes were identified using fold change (FC) > |1.4|, FDR < 0.05.

# Data Availability

The global gene expression data are deposited in Gene Expression Omnibus by following MIAME guidelines and can be accessed with the dataset identification number GSE225283.

# Supplementary Information

# Acknowledgements

Authors thank Dr. Rainer Fürbaß for gifting the aromatase antibody and Dr. Chitneedi for helping to generate the circular plot. Authors acknowledge Christian Plinski, Veronica Schreiter, Maren Anders, Ursula Antkewitz and Rodewald Swanhild for their technical support. This work is funded by Deutsche Forschungsgemeinschaft (DFG; Grant No. BA 6909/1-1) to VS Baddela.

## Author Contributions

VS Baddela: conceptualization, formal analysis, funding acquisition, investigation, visualization, and writing—original draft, review, and editing.

M Michaelis: data curation, formal analysis, and investigation.

X Tao: data curation, formal analysis, and investigation.

D Koczan: investigation.

J Vanselow: conceptualization, data curation, and writing—review and editing.

## Conflict of Interest Statement

The authors declare that they have no conflict of interest.

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
