## [Reviewer comments · Life Science Alliance]

Life Science Alliance

ERK1/2-SOX9/FOXL2 axis regulates ovarian steroidogenesis and favors the follicular-luteal transition

Vijay Baddela, Marten Michaelis, Xuelian Tao, Dirk Koczan, and Jens Vanselow

DOI: <https://doi.org/10.26508/lsa.202302100>

Corresponding author(s): *Vijay Baddela, Research Institute for Farm Animal Biology*

Review Timeline:	Submission Date:	2023-04-17
	Editorial Decision:	2023-05-26
	Revision Received:	2023-06-28
	Editorial Decision:	2023-07-14
	Revision Received:	2023-07-19
	Accepted:	2023-07-24

Transaction Report:

May 26, 2023

Re: Life Science Alliance manuscript #LSA-2023-02100-T

Dr. Vijay Simha Baddela
Research Institute for Farm Animal Biology
Reproductive Biology
Wilhelm Stahl Allee 2
Dummerstorf, MV 18196
Germany

Dear Dr. Baddela,

Thank you for submitting your manuscript entitled "ERK1/2-SOX9/FOXL2 axis regulates ovarian steroidogenesis and favors the follicular-luteal transition" to Life Science Alliance. The manuscript was assessed by expert reviewers, whose comments are appended to this letter. We invite you to submit a revised manuscript addressing the Reviewer comments.

Thank you for this interesting contribution to Life Science Alliance. We are looking forward to receiving your revised manuscript.

Sincerely,

B. MANUSCRIPT ORGANIZATION AND FORMATTING:

Reviewer #1 (Comments to the Authors (Required)):

Title: ERK1/2-SOX9/FOXL2 axis regulates ovarian steroidogenesis and favors the follicular-luteal transition.

Authors: BaddelaVanselow

Review: This manuscript provides detailed analyses of the impact of ERK signaling in primary granulosa cells obtained from small follicles of bovine ovaries. The investigators use multiple assays and approaches to define the impact of ERK signaling on granulosa cell steroidogenesis that occurs during the transition of granulosa cells to a luteal-like phenotype. The methods are clearly and carefully described. Likewise the presentation of results is elegant. The conclusion (except for the last sentence) is justified by the results presented.

Specific comments:

Figure 1 documents that inhibition of ERK signaling using specific inhibitors of ERK phosphorylation results in 1) markedly increased expression of aromatase (CYP19A1) mRNA and protein and estradiol production and 2) marginal but significantly decreased levels of STAR mRNA and protein and progesterone production without changes in CYP11A1 levels.

Figure 2 documents that reducing ERK expression using specific ERK Gapmers leads to increased aromatase mRNA and estradiol and decreased STAR mRNA and progesterone. Specific activation of ERK with TPA causes the reverse responses by nearly obliterating aromatase mRNA, protein and estradiol levels but curiously had no effect on STAR mRNA or progesterone levels.

Figure 3 provides evidence for changes in cell proliferation as well as global changes in the expression of specific genes related to FSH, LH and IGF1 signaling, including CYP19A1, ESR1, FOXL2, SOX9 and others.

Figures 4/5 provide evidence that FOXL2 and SOX9 play pivotal, and apposing, roles in mediating ERK signaling and steroidogenesis in granulosa cells. Specifically, SOX9 antagonizes the expression of CYP19A1 and estradiol biosynthesis whereas SOX9 promotes expression of STAR and progesterone. SOX9 appears to be a major regulator of estradiol to progesterone transition during "luteinization" in this model.

Other considerations:

Additional markers of "luteinization" would be desirable. While proliferation may be one marker of granulosa cell activity during very early stages of luteinization, granulosa cells actually cease dividing as mature luteal cells. All of the assays are done for relatively short duration in culture. Do corpora lutea of cattle express SOX9? Does SOX9 actually increase in recently ovulated normal follicles? Do theca cells have an impact?

Luteal cells are also morphologically enlarged with specific mitochondrial features. What are the morphological features of the cells analyzed in Figure 5? Have they enlarged? Are the mitochondria different? For how long is SOX9 expressed?

What happens if FOXL2 expression is reduced by FOXL2 Gapmers?

The investigators might consider analyzing expression of INSL3 which appears on the RNA-seq analysis. This is a key marker of luteal cells and especially progesterone producing cells.

Lines 48-50: FSH enhances aromatase expression but LH mediated androgen production by theca cells is required for estradiol biosynthesis. Estradiol at elevated levels induces a "positive" feedback loop.

Lines 401-402. How about the role of INSL3 that is a strong marker of steroidogenic luteal cells.

Lines 406-412: This is a long sentence!

Lines 432-434: why not use EGF or LH or something more physiological? What are FSHR and LHCGR levels in these cells? Do they change?

Concluding sentence: This a bit of a leap. PCOS is largely a theca cell, LH and androgen disorder with premature arrest of growing follicles. Is there evidence for altered ERK/FOXL2/SOX9 in PCOS ovaries??!!

Reviewer #2 (Comments to the Authors (Required)):

Vijay Simha Baddela et al. studied the regulation of ERK1/2 cascade by utilizing bovine primary granulosa cell culture system. They highlighted the role of ERK1/2 in augmenting granulosa cell differentiation and promoting ovarian steroidogenesis. ERK1/2 downregulated FOXL2 but upregulated SOX9 expression, eventually ceasing aromatase and estradiol production while increasing progesterone. Although Vijay Simha Baddela et al. provided solid and high-quality results, this manuscript could be improved by addressing the following comments.

Major comments:

1. The template of this manuscript should be revised based on the guidance of LSA. A complex discussion session should be included. The authors might want to describe only objective findings in the results without discussing them.
2. What are the novelties of this research, as the role of ERK1/2 has been thoroughly studied and reported?
3. How did the authors decide on the concentrations of PD98059 or U0126? Have gradient concentrations been tested in-house?
4. What was the cell viability and activity after being treated with PD98059 or TPA?
5. How about the gene expression profile in ERK1 KD granulosa cells? Have the authors analyzed gene regulation profiles of ERK KD granulosa cells in parallel with PD98059 treated granulosa cells?
6. Have the authors measured LH, FSH, estradiol, and progesterone expression levels in the conditioned media by ELISA? It could be better to include ELISA results to reflect the hormone profile.
7. What would be the potential application of the key findings from this study? How likely are the findings of this article to be applied clinically or in agriculture?
8. Have the authors isolated luteinized granulosa cells from bovine corpus luteum to compare with the TPA-activated granulosa cells?

Minor comments:

1. Please include proper X axis titles/indications in each figure.
2. Some of the words/labels are masked/missing in the figures.

Reviewer #3 (Comments to the Authors (Required)):

The manuscript entitled "ERK1/2-SOX9/FOXL2 axis regulates ovarian steroidogenesis and favors the follicular-luteal transition" by Baddela et al., is a very insightful research regarding the molecular process pertaining to the shift of estradiol production to progesterone production during follicular luteal transition. All the essential experiments were performed well with utmost precision to confirm the involvement of ERK1/2-SOX9/FOXL2 axis for the estradiol to progesterone production. The manuscript is also written very well. Therefore, the manuscript is recommended for publishing.

Reviewers' comments**Reviewer #1**

Review: This manuscript provides detailed analyses of the impact of ERK signaling in primary granulosa cells obtained from small follicles of bovine ovaries. The investigators use multiple assays and approaches to define the impact of ERK signaling on granulosa cell steroidogenesis that occurs during the transition of granulosa cells to a luteal-like phenotype. The methods are clearly and carefully described. Likewise, the presentation of results is elegant. The conclusion (except for the last sentence) is justified by the results presented.

Response: *Dear reviewer, thank you for sparing time and presenting insightful suggestions and comments on the manuscript, and appreciating the manuscript for its strengths. We addressed your concerns and edited the conclusion to avoid overinterpretation of the data in the revised manuscript.*

Specific comments:

Comment 1. Figure 1 documents that inhibition of ERK signaling using specific inhibitors of ERK phosphorylation results in 1) markedly increased expression of aromatase (CYP19A1) mRNA and protein and estradiol production and 2) marginal but significantly decreased levels of STAR mRNA and protein and progesterone production without changes in CYP11A1 levels.

Response: *Thank you for the summary of Figure 1.*

Comment 2. Figure 2 documents that reducing ERK expression using specific ERK Gapmers leads to increased aromatase mRNA and estradiol and decreased STAR mRNA and progesterone. Specific activation of ERK with TPA causes the reverse responses by nearly obliterating aromatase mRNA, protein and estradiol levels but curiously had no effect on STAR mRNA or progesterone levels.

Response: *Thank you for the summary of Figure 2.*

Comment 3. Figure 3 provides evidence for changes in cell proliferation as well as global changes in the expression of specific genes related to FSH, LH and IGF1 signaling, including CYP19A1, ESR1, FOXL2, SOX9 and others.

Response: *Thank you for the summary of Figure 3.*

Comment 4. Figures 4/5 provide evidence that FOXL2 and SOX9 play pivotal, and apposing, roles in mediating ERK signaling and steroidogenesis in granulosa cells. Specifically, SOX9 antagonizes the expression of CYP19A1 and estradiol biosynthesis whereas SOX9 promotes expression of STAR and progesterone. SOX9 appears to be a major regulator of estradiol to progesterone transition during "luteinization" in this model.

Response: Thank you for the summary of Figures 4 and 5.

Other considerations:

Comment 5. Additional markers of "luteinization" would be desirable. While proliferation may be one marker of granulosa cell activity during very early stages of luteinization, granulosa cells actually cease dividing as mature luteal cells. All of the assays are done for a relatively short duration in culture. Do corpora lutea of cattle express SOX9? Does SOX9 actually increase in recently ovulated normal follicles? Do theca cells have an impact?

Response: Thank you for the suggestions and comments.

During folliculogenesis, LH surge induces dramatic changes in gene expression profile via activating multiple signaling pathways. One of the prominent functional changes induced by LH is the inhibition of estradiol production and induction of progesterone production. This change in steroidogenesis takes place while granulosa cells are still within the ovarian follicle before ovulation but post-LH surge (Nimz et al. 2009; Christenson et al. 2013).

With the theoretical background of increased ERK signaling in granulosa cells of post-LH follicles necessary for ovulation, we wonder whether ERK also impacts steroidogenesis. Here, using multiple approaches, we identify that ERK negatively regulates estradiol production and positively regulates progesterone production in granulosa cells. However, the manuscript does not advise the complete transformation of granulosa cells into luteal cells by ERK signaling.

However, based on the suggestion, we analyzed the expression of two additional genes, FSHR and PTX3. Expression of FSHR and PTX3 were found to be decreased and increased, respectively, after LH surge (Christenson et al. 2013). The present analysis showed that inhibition of ERK signaling induced FSHR and inhibited PTX3 gene expression in granulosa cells, indicating activation of ERK signaling might be responsible for the LH-induced regulation of FSHR and PTX3 genes along with other genes mentioned in the manuscript. This additional data can be found in supplementary information 1 of the manuscript.

As rightfully pointed out by the reviewer, many reports suggest that cell proliferation is nearly ceased in mature luteal cells. However, we believe that the treatments of the present study did not induce the formation of mature luteal cells but mimicked the post-LH stage granulosa cells of the ovarian follicle. Given the fact that cell signaling and protein phosphorylation events are very dynamic and short-term, we believe it is imperative to carry out experiments for a short duration to achieve consistent results.

SOX9 expression was previously identified in bovine corpus luteum (Toyokawa et al. 2011). However, based on the comment, we have verified SOX9 expression in mid-cycle (day 10-12) corpus luteum samples and found a positive PCR signal. As already mentioned in the manuscript, expression of SOX9 was found to increase in granulosa cells collected from the ovulated follicles of human patients after recombinant LH treatment; however, no information is available in recently ovulated follicular granulosa cells under normal circumstances and needs future studies. We have not studied the impact of ERK signaling on thecal cells in the present study. It was shown that ERK signaling inhibits CYP17A1 production and regulates androgen production in thecal cells (Nelson-Degrave et al. 2005). Therefore, it would be interesting to pursue functional studies in thecal cells to further understand ERK signaling.

Comment 6. Luteal cells are also morphologically enlarged with specific mitochondrial features. What are the morphological features of the cells analyzed in Figure 5? Have they enlarged? Are the mitochondria different? For how long is SOX9 expressed?

Response: We agree with the reviewer that the mature luteal cells are morphologically distinct and enlarged compared to their predecessor granulosa cells/theca cells. However, we did not detect significant morphological changes among cells with low or high ERK phosphorylation levels. Here we present the photomicrographs and flow cytometry forward light scatter values, which indicate the size of the cells, showing no significant changes in the cell morphology/size upon manipulating ERK signaling in granulosa cells. This again suggests that granulosa cells are not differentiated into mature luteal cells under present culture conditions. The cell morphology and size data are now presented in supplementary information 1 of the manuscript.

Cell morphology and size: A) Representative microphotographs of cultured cells in different treatments. The scale bar size is approximately 100 μ M. B) Representative contour plots indicating forward scattering of light in a flow cytometer, a measure of cell size. C) Quantification of forward scattering data (n=4).

Dear reviewer, we did not analyze the mitochondrial structure and function, as it will deviate the present study's focus from ERK signaling-induced gene regulation of steroidogenesis in granulosa cells. However, we think it would be an excellent idea for future studies on the role of the ERK-signaling axes on mitochondrial structure and function in steroidogenic cells. Regarding SOX9 expression, we infer from our studies that SOX9 expression is very responsive to ERK phosphorylation, as PD98059 and TPA treatments inhibited and induced the SOX9 expression, respectively (Figure 4 of the manuscript). We have measured SOX9 mRNA and protein expressions at 48 hr and 96 hr after inducing ERK phosphorylation. However, we did not analyze how long SOX9 mRNA or protein can be expressed in these cells. We think this is subject to ERK activity and need further dedicated investigations on how long ERK induces SOX9.

Comment 7. What happens if FOXL2 expression is reduced by FOXL2 Gapmers?

Response: *Dear reviewer, we have not performed FOXL2 gene silencing in the present study. However, earlier studies in mice granulosa cells showed that FOXL2 gene silencing leads to a decrease in CYP19A1 expression and diminished estradiol production (Herman et al. 2021). A detailed description of FOXL2 effects in granulosa cells has been provided in the manuscript.*

Comment 8. The investigators might consider analyzing the expression of INSL3 which appears in the RNA-seq analysis. This is a key marker of luteal cells and especially progesterone-producing cells.

Response: *Thank you for the suggestion. We have now analyzed INSL3 expression in granulosa cells and found that INSL3 expression is inhibited by ERK signaling. We*

agree with you that INSL3 is enriched in thecal cells of the ovarian follicle and luteal cells of the corpus luteum (Balvers et al. 1998; Hanna et al. 2010; Pitia et al. 2021).

Interestingly, studies indicate that LH surge decreases INSL3 mRNA expression in thecal cells and its protein level in follicular fluid (Satchell et al. 2013). Therefore, the downregulation of INSL3 expression by ERK in granulosa cells is comparable to INSL3 downregulation in thecal cells after LH surge. However, post-LH staged granulosa/theca cells are morphologically and functionally distinct compared to mature luteal cells. We believe it needs further investigation to understand the role of INSL3 in post-LH staged cells and mature luteal cells. This data can be found in supplementary information 1 of the manuscript. Thank you.

ERK-induced gene expression regulation: mRNA expression quantification INSL3 gene in control and PD98059 treated cells. The Left Y axis is the data from qPCR analysis (n=6) and the right Y axis is the data from transcriptome analysis

Comment 9. Lines 48-50: FSH enhances aromatase expression but LH mediated androgen production by theca cells is required for estradiol biosynthesis. Estradiol at elevated levels induces a "positive" feedback loop.

Response: Thank you for the correction. The sentence is revised accordingly.

Comment 10. Lines 401-402. How about the role of INSL3 that is a strong marker of steroidogenic luteal cells.

Response: Dear reviewer, this comment is addressed in our response to comment 8. Thank you.

Comment 11. Lines 406-412: This is a long sentence!

Response: Thank you for spotting the correction. These lines are now reorganized in shorter sentences in the revised manuscript lines 254-259.

Comment 12. Lines 432-434: why not use EGF or LH or something more physiological? What are FSHR and LHCGR levels in these cells? Do they change?

Response:

We agree that applying EGF or LH would be more physiological. However, the granulosa cells used in this model were collected from the small to medium-sized follicles (< 6mm) and are not responsive to LH as the expression of LHCGR is far lower than FSHR in these cells. This information is now provided in the manuscript line 348-349. As shown in the figure, the expression of FSHR was increased after inhibition of ERK signaling, whereas LHCGR expression was not regulated under the present conditions.

ERK-induced gene expression regulation: mRNA expression of FSHR and LHCGR in >6mm follicular bovine granulosa cells in control and PD98059 treated cells in transcriptome data.

Comment 13. Concluding sentence: This a bit of a leap. PCOS is largely a theca cell, LH and androgen disorder with premature arrest of growing follicles. Is there evidence for altered ERK/FOXL2/SOX9 in PCOS ovaries??!!

We thank the reviewer for pointing out this sentence for correction. We agree that PCOS is largely a disorder of excess androgen production by theca cells. To the best of our knowledge, this is the first report on ERK-SOX9/FOXL2 axis in ovarian granulosa cells and its involvement in steroidogenesis. We have removed the last sentence from the revised manuscript to avoid over-interpreting the data.

Dear reviewer, We are delighted to address the comments and make amendments in the manuscript accordingly to help better shape the manuscript. Thanks a lot!

Reviewer #2

Vijay Simha Baddela et al. studied the regulation of ERK1/2 cascade by utilizing bovine primary granulosa cell culture system. They highlighted the role of ERK1/2 in augmenting granulosa cell differentiation and promoting ovarian steroidogenesis. ERK1/2 downregulated FOXL2 but upregulated SOX9 expression, eventually ceasing aromatase and estradiol production while increasing progesterone. Although Vijay Simha Baddela et al. provided solid and high-quality results, this manuscript could be improved by addressing the following comments.

Response: *Dear reviewer, thank you for sparing time for presenting your summary and suggestions on the manuscript, and appreciating the manuscript for its strengths. We have addressed all your comments in the revised manuscript and the amendments, wherever incorporated indicated in the response to corresponding comment.*

Major comments:

Comment 1. The template of this manuscript should be revised based on the guidance of LSA. A complex discussion section should be included. The authors might want to describe only objective findings in the results without discussing them.

Response: *Dear reviewer, thank you for your suggestion. We have now ensured the structure of the manuscript matches up with LSA guidelines. Especially, the manuscript is now reorganized by placing the materials and methods section after the results and discussion section. However, we wish to have a combined result and discussion section, which is within the guidelines of LSA for shorter articles since the present manuscript is prepared in a short article format. We believe that the combined results and discussion section allow the readers to have a smooth reading of the manuscript and simultaneous interpretation of the data.*

Comment 2. What are the novelties of this research, as the role of ERK1/2 has been thoroughly studied and reported?

Response: *As rightfully pointed out by the reviewer, ERK is one of the well-documented signaling pathways in different cell types and species. Most of the earlier studies on ERK in the ovary aimed to describe its role in ovulation. In this study, using multiple experimental approaches, we show that induction of ERK signaling could contribute to the immediate downregulation of estradiol and upregulation of progesterone production, a phenomenon of post-LH follicles. Further, we present genome-wide gene expression changes induced by ERK signaling in granulosa cells that would be a resource point for future research on ERK signaling in granulosa cells. Most importantly, we identify a novel signaling axis of ERK i.e. ERK-SOX9/FOXL2, as a critical regulator of steroidogenesis, which we suggest to be*

taken into account while addressing ovarian steroidogenesis disorders in humans and animals. Thank you.

Comment 3. How did the authors decide on the concentrations of PD98059 or U0126? Have gradient concentrations been tested in-house?

Response: Dear reviewer, we have used 50 μM of PD98059 and 10 μM of U0126 for inhibiting the ERK phosphorylation in the present study. These concentrations were selected based on previous publications (Cottom et al. 2003; Puttabyatappa et al. 2013; Donaubauer and Hunzicker-Dunn 2016) and recommendations of the manufacturer (cell signaling technology, USA. <https://www.cellsignal.com>) but not based on the in-house analysis. This information is already mentioned in the manuscript. Thank you.

Comment 4. What was the cell viability and activity after being treated with PD98059 or TPA?

Response: Dear reviewer, Thank you for this helpful comment. We found no differences in the cell viability/death status in different treatments, as shown below. For this analysis, cultured cells were detached from the plate, stained with propidium iodide, and analyzed using a flow cytometer. This data is now added to supplementary information 1.

Cell viability: A) Representative scatter diagrams of viable (Green; PI negative) and dead (red; PI positive) cells in different treatments. B) Quantification of cell viability treatments (n=4).

Comment 5. How about the gene expression profile in ERK1 KD granulosa cells? Have the authors analyzed gene regulation profiles of ERK KD granulosa cells in parallel with PD98059 treated granulosa cells?

Response: Dear reviewer, we undertook the ERK gene silencing experiments (Figure 2) to validate the effects of ERK inhibitors, and we found nearly similar results (Figure 1). However, we intentionally did not consider the global gene expression profile analysis in ERK knockdown cells since ERK is mainly subjected to changes in

its phosphorylation status rather than its expression changes in granulosa cells. Thank you.

Comment 6. Have the authors measured LH, FSH, estradiol, and progesterone expression levels in the conditioned media by ELISA? It could be better to include ELISA results to reflect the hormone profile.

Response: *We have analyzed the estradiol and progesterone levels in the media using radioimmunoassays (RIA) as described in the manuscript (Lines 367-383). We RIA is considered to be more sensitive compared to ELISA assays. However, We did not analyze the LH and FSH levels since these hormones will be coming from the anterior pituitary but not synthesized in the granulosa cells.*

Comment 7. What would be the potential application of the key findings from this study? How likely are the findings of this article to be applied clinically or in agriculture?

Response: *Dear reviewer, it's always challenging to find the direct application of basic science data like this. Nevertheless, we could think of at least two implications of the data. Our data strongly argue that ERK is a negative regulator of estradiol production in ovarian granulosa cells. Since estradiol levels are very important for maintaining healthy bones, skin, and also brain function, the present data could be a foundation for future studies in humans, especially for addressing low estrogen-driven disorders using ERK inhibitors. Secondly, it lays a basis for verifying whether the in vivo corpus luteum function can be improved by ERK activators since luteal insufficiency is a major cause of early embryonic death in cows. We now include these lines in the manuscript as future implications and placed above the conclusions. Thank you.*

Comment 8. Have the authors isolated luteinized granulosa cells from bovine corpus luteum to compare with the TPA-activated granulosa cells?

Response: *Dear reviewer, we haven't performed a comparative analysis of luteal cells of corpus luteum and TPA-treated granulosa cells as this study is primarily intended to analyze molecular changes induced by ERK signaling in granulosa cells to understand the post-LH staged cells. During folliculogenesis, LH surge induces dramatic changes in gene expression profile via activating multiple signaling pathways. One of the prominent changes induced by LH is the inhibition of estradiol production and induction of progesterone production. This steroidogenesis switch occurs while granulosa cells are still within the ovarian follicle before ovulation converting into luteal cells (Nimz et al. 2010; Christenson et al. 2013). With the theoretical background of increased ERK signaling in granulosa cells of post-LH follicles, here, using multiple approaches, we identify that ERK negatively regulates estradiol production and positively regulates progesterone production in granulosa*

cells. However, the manuscript does not advise for full transformation of granulosa cells into luteal cells by ERK signaling.

Minor comments:

Comment 1. Please include proper X axis titles/indications in each figure.

Comment 2. Some of the words/labels are masked/missing in the figures.

Response: *Dear reviewer, thank you for your concern. We have verified all the figures carefully based on your comment. We want to bring to your notice that, if the labels are the same for both lower and upper panel graphs in a figure, we chose to provide the X-axis labels only for lower panel graphs to spare space and avoid the repetition of labels. However, we now added labels in Figure 2 for western image 2O in the revised manuscript. Thank you.*

We are delighted to address the comments and make amendments in the manuscript accordingly to help better shape the manuscript. Thank you for the help once again!

Reviewer #3 (Comments to the Authors (Required)):

The manuscript entitled "ERK1/2-SOX9/FOXL2 axis regulates ovarian steroidogenesis and favors the follicular-luteal transition" by Baddela et al., is a very insightful research regarding the molecular process pertaining to the shift of estradiol production to progesterone production during follicular luteal transition. All the essential experiments were performed well with utmost precision to confirm the involvement of ERK1/2-SOX9/FOXL2 axis for the estradiol to progesterone production. The manuscript is also written very well. Therefore, the manuscript is recommended for publishing.

Response: *Dear reviewer, thank you for sparing time and presenting your view on the manuscript. We feel very happy to find your appreciation on the quality of the manuscript and recommendation for publication.*

Thank you very much!

- Balvers M, Spiess A-N, Domagalski R, Hunt N, Kilic E, Mukhopadhyay A, Hanks E, Charlton H, Ivell R. 1998. Relaxin-like factor expression as a marker of differentiation in the mouse testis and ovary. *Endocrinology* **139**: 2960-2970.
- Christenson LK, Gunewardena S, Hong X, Spitschak M, Baufeld A, Vanselow J. 2013. Research resource: preovulatory LH surge effects on follicular theca and granulosa transcriptomes. *Mol Endocrinol* **27**: 1153-1171.
- Cottom J, Salvador LM, Maizels ET, Reierstad S, Park Y, Carr DW, Davare MA, Hell JW, Palmer SS, Dent P et al. 2003. Follicle-stimulating hormone activates extracellular signal-regulated kinase but not extracellular signal-regulated kinase kinase through a 100-kDa phosphotyrosine phosphatase. *J Biol Chem* **278**: 7167-7179.
- Donaubauer EM, Hunzicker-Dunn ME. 2016. Extracellular signal-regulated kinase (ERK)-dependent phosphorylation of Y-box-binding protein 1 (YB-1) enhances gene expression in granulosa cells in response to follicle-stimulating hormone (FSH). *Journal of Biological Chemistry* **291**: 12145-12160.
- Hanna CB, Yao S, Patta MC, Jensen JT, Wu X. 2010. Expression of insulin-like 3 (INSL3) and differential splicing of its receptor in the ovary of rhesus macaques. *Reproductive Biology and Endocrinology* **8**: 1-9.
- Herman L, Legois B, Todeschini AL, Veitia RA. 2021. Genomic exploration of the targets of FOXL2 and ESR2 unveils their implication in cell migration, invasion, and adhesion. *The FASEB Journal* **35**: e21355.
- Nelson-Degrave VL, Wickenheisser JK, Hendricks KL, Asano T, Fujishiro M, Legro RS, Kimball SR, Strauss III JF, McAllister JM. 2005. Alterations in mitogen-activated protein kinase kinase and extracellular regulated kinase signaling in theca cells contribute to excessive androgen production in polycystic ovary syndrome. *Molecular Endocrinology* **19**: 379-390.
- Nimz M, Spitschak M, Fuerbass R, Vanselow J. 2010. The preovulatory luteinizing hormone surge is followed by downregulation of *CYP19A1*, *HSD3B1* and *CYP17A1* and chromatin condensation of the corresponding promoters in bovine follicles. *Molecular reproduction and development* **77**: 1040-1048.
- Nimz M, Spitschak M, Schneider F, Fürbass R, Vanselow J. 2009. Down-regulation of genes encoding steroidogenic enzymes and hormone receptors in late preovulatory follicles of the cow coincides with an accumulation of intrafollicular steroids. *Domest Anim Endocrinol* **37**: 45-54.
- Pitia AM, Minagawa I, Abe Y, Kizaki K, Hamano K-i, Sasada H, Hashizume K, Kohsaka T. 2021. Evidence for existence of insulin-like factor 3 (INSL3) hormone-receptor system in the ovarian corpus luteum and extra-ovarian reproductive organs during pregnancy in goats. *Cell and Tissue Research* **385**: 173-189.
- Puttabyatappa M, Brogan RS, Vandervoort CA, Chaffin CL. 2013. EGF-like ligands mediate progesterone's anti-apoptotic action on macaque granulosa cells. *Biol Reprod* **88**: 18.
- Satchell L, Glistler C, Bleach EC, Glencross RG, Bicknell AB, Dai Y, Anand-Ivell R, Ivell R, Knight PG. 2013. Ovarian expression of insulin-like peptide 3 (INSL3) and its receptor (RXFP2) during development of bovine antral follicles and corpora lutea and measurement of circulating INSL3 levels during synchronized estrous cycles. *Endocrinology* **154**: 1897-1906.
- Toyokawa K, Liu W, Pate JL. 2011. Expression and Regulation of PRAME in the Bovine Corpus Luteum. Oxford University Press.

July 14, 2023

RE: Life Science Alliance Manuscript #LSA-2023-02100-TR

Dr. Vijay Simha Baddela
Research Institute for Farm Animal Biology
Reproductive Biology
Wilhelm Stahl Allee 2
Dummerstorf, MV 18196
Germany

Dear Dr. Baddela,

Thank you for submitting your revised manuscript entitled "ERK1/2-SOX9/FOXL2 axis regulates ovarian steroidogenesis and favors the follicular-luteal transition". We would be happy to publish your paper in Life Science Alliance pending final revisions necessary to meet our formatting guidelines.

- please note that supplementary figures should be uploaded individually like the main figures
- each supplementary figure should be labeled starting with "Figure S1..." and cited accordingly in the main manuscript text
- please upload your Tables in editable .doc or excel format; -Tables should be numbered consecutively with Arabic numerals (1, 2, 3, 4), i.e., S1, S2, S3... in case of supplementary tables
- please add the Twitter handle of your host institute/organization as well as your own or/and one of the authors in our system
- please incorporate any points from the Conclusion section into the Discussion. We only allow a Discussion section
- please label the references section as "References."
- please add your main, supplementary figure, and table legends to the main manuscript text after the references section
- please add an Author Contributions section to your main manuscript text
- please add callouts for Figures 2E and 5I to your main manuscript text
- please cite your tables in the manuscript text accordingly
- the Conclusion section should be incorporated into the Results and Discussion section

Figure checks:

- the Western blots are created digitally. These are just based upon the quantitation that is presented next to each blot, correct? Does the digitally-created blot add anything of significance?
- scale bars in Figure 4H are hard to read

A. FINAL FILES:

-- Summary blurb (enter in submission system): A short text summarizing in a single sentence the study (max. 200 characters including spaces). This text is used in conjunction with the titles of papers, hence should be informative and complementary to

the title. It should describe the context and significance of the findings for a general readership; it should be written in the present tense and refer to the work in the third person. Author names should not be mentioned.

B. MANUSCRIPT ORGANIZATION AND FORMATTING:

Sincerely,

Reviewer #1 (Comments to the Authors (Required)):

The authors have carefully addressed all concerns and comments.
The revised article is an excellent and elegant presentation of the data provided.

Please accept this manuscript for publication

Reviewer #2 (Comments to the Authors (Required)):

The authors have thoughtfully addressed all the questions/concerns and revised the manuscript accordingly. The reviewer has no more questions regarding the new version. Therefore, the revised manuscript is recommended for publishing in LSA.

Final remarks

Remark 1: please note that supplementary figures should be uploaded individually like the main figures

Response: *Done, Thank you.*

Remark 2: -each supplementary figure should be labeled starting with "Figure S1..." and cited accordingly in the main manuscript text

Response: *Done, Thank you.*

Remark 3: -please upload your Tables in editable .doc or excel format; -Tables should be numbered consecutively with Arabic numerals (1, 2, 3, 4), i.e., S1, S2, S3... in case of supplementary tables

Response: *Done, Thank you.*

Remark 4: -please add the Twitter handle of your host institute/organization as well as your own or/and one of the authors in our system

Response: *Done, Thank you.*

Remark 5: -please incorporate any points from the Conclusion section into the Discussion. We only allow a Discussion section

Response: *Done, Thank you.*

Remark 6: -please label the references section as "References."

Response: *Done, Thank you.*

Remark 7: -please add your main, supplementary figure, and table legends to the main manuscript text after the references section

Response: *Done, Thank you.*

Remark 8: -please add an Author Contributions section to your main manuscript text

Response: *Done, Thank you.*

Remark 9: -please add callouts for Figures 2E and 5I to your main manuscript text

Response: *Done, Thank you.*

Remark 10: -please cite your tables in the manuscript text accordingly

Response: *Done, Thank you.*

Remark 11: -the Conclusion section should be incorporated into the Results and Discussion section

Response: *Done, Thank you.*

Figure checks:

Remark 12: -the Western blots are created digitally. These are just based upon the quantitation that is presented next to each blot, correct? Does the digitally-created blot add anything of significance?

Response:

Dear Editor, Western analyses were performed in a capillary electrophoresis system using Protein Simple's Wes instrument as indicated in the manuscript. Detection of proteins in the Wes system is conceptually the same as conventional Western blots. Wes system performs protein fractionation in capillaries and immobilizes the fractionated proteins into the wall of capillaries by UV crosslink. The chemiluminescence detection is then performed in the capillaries using the given primary antibody and secondary antibody module. It is measured by the inbuilt camera detector throughout the length of the capillary and generates an electropherogram which is digitally recreated as a gel image for each run. The luminescence snapshot is evaluated and quantified by software (Compass for SW, version 6.0.0).

Therefore, the presented Western images are based on the electropherogram of wes run and we believe they add significance similar to a conventional Western blot by giving details about the specificity of signal and molecular weight details.

Thank you.

Remark 13: -scale bars in Figure 4H are hard to read

Response: *The scale bars are revised. Thank you.*

July 24, 2023

RE: Life Science Alliance Manuscript #LSA-2023-02100-TRR

Dr. Vijay Simha Baddela
Research Institute for Farm Animal Biology
Reproductive Biology
Wilhelm Stahl Allee 2
FBN
Dummerstorf, MV 18196
Germany

Dear Dr. Baddela,

Thank you for submitting your Research Article entitled "ERK1/2-SOX9/FOXL2 axis regulates ovarian steroidogenesis and favors the follicular-luteal transition". It is a pleasure to let you know that your manuscript is now accepted for publication in Life Science Alliance. Congratulations on this interesting work.

DISTRIBUTION OF MATERIALS:

Again, congratulations on a very nice paper. I hope you found the review process to be constructive and are pleased with how the manuscript was handled editorially. We look forward to future exciting submissions from your lab.

Sincerely,
